# Residual Tumor Patterns of Breast Cancer on MRI after Neo-Adjuvant Chemotherapy: Impact on Clinicopathologic Factors and Prognosis

**DOI:** 10.3390/diagnostics12102294

**Published:** 2022-09-23

**Authors:** Yoon Jin Cha, Na Lae Eun, Dooreh Kim, Soong June Bae, Sung Gwe Ahn, Joon Jeong, Woo-Chan Park, Yangkyu Lee, Chang Ik Yoon

**Affiliations:** 1Department of Pathology, Gangnam Severance Hospital, Yonsei University College of Medicine, Seoul 06273, Korea; 2Institute of Breast Cancer Precision Medicine, Yonsei University College of Medicine, Seoul 06273, Korea; 3Department of Radiology, Gangnam Severance Hospital, Yonsei University College of Medicine, Seoul 06273, Korea; 4Division of Breast Surgery, Department of Surgery, Seoul St. Mary’s Hospital, College of Medicine, The Catholic University of Seoul, Seoul 06591, Korea; 5Department of Surgery, Gangnam Severance Hospital, Yonsei University College of Medicine, Seoul 06273, Korea

**Keywords:** breast neoplasms, neoadjuvant therapy, magnetic resonance imaging, prognosis

## Abstract

(1) Background: Residual breast cancer after neoadjuvant chemotherapy (NAC) could have a variable image pattern on a follow-up breast magnetic resonance image (MRI). In this study, we compared the clinical outcome of breast cancer patients with different residual tumor patterns (RTP) on a breast MRI after NAC. (2) Methods: A total of 91 patients with breast cancer who received NAC and subsequent curative surgery were selected. All included patient had residual breast cancer after NAC and showed a partial response on a breast MRI. Pre- and post-treatment were reviewed by an experienced radiologist to evaluate different RTP, and classified into two groups: concentric and scattered patterns. The clinicopathologic parameters and survival outcomes [recurrence-free survival (RFS) and distant metastasis-free survival (DMFS)] were analyzed according to different RTP. (3) Results: Patients with a scattered pattern had a larger extent of pre-treated non-mass enhancement and more frequently received total mastectomy. With a median follow-up period of 37 months, RTP were not significantly associated with RFS or DMFS. (4) Conclusions: In the patients with residual breast cancer after NAC, RTP on an MRI had no effect on the patients’ clinical outcome. The curative resection of the tumor bed and securing the negative resection margins appear to be important in the treatment of patients with residual breast cancer after NAC.

## 1. Introduction

Neoadjuvant chemotherapy (NAC) is the standard treatment for patients with locally advanced breast cancer, and it is increasingly being administered to patients with early breast cancer as well. By decreasing the tumor size and reducing nodal metastasis, NAC may enable patients to undergo breast conserving surgery or to avoid axillary lymph node dissection. Achieving a pathologic complete response (pCR) after NAC significantly improves survival, particularly in patients with human epidermal growth factor receptor-2 (HER2) positive and triple negative breast cancer (TNBC) [1].

After NAC, the treatment response of patients with residual breast cancer can be classified into two patterns on magnetic resonance imaging (MRI) [2]:(1)Concentric shrinkage pattern that finally achieves a smaller tumor size than that of the initial tumor area.(2)Diffusely scattered pattern with maintenance of the initial tumor area.

Previous studies have shown that different residual tumor patterns (RTP) on an MRI are associated with the distributed pattern of the primary tumor and molecular subtypes—a concentric shrinkage pattern was frequently observed in TNBC [3,4]. In patients with low-grade luminal breast cancer, a concentric shrinkage pattern was associated with superior disease-free survival (DFS) and overall survival (OS) [5]. To date, the Residual Cancer Burden (RCB) score and RCB index are commonly used to evaluate the treatment response on NAC in breast cancer [6,7]. RCB classes are now established prognostic indicators that effectively stratify the patients after long-term follow up [8]. However, whether different RTPs on an MRI affect the prognosis of patients with residual breast cancer after NAC remains unexplored. In the present study, we aimed to evaluate the clinical outcome of patients with breast cancer after NAC based on the different RTPs on the MRI.

## 2. Materials and Methods

### 2.1. Study Population

We retrospectively collected electronic medical data from patients treated with NAC before curative surgery for breast cancer at the Gangnam Severance Hospital in Seoul, Korea, from January 2009 to December 2017. These patients were treated according to standard protocols. A total of 2987 patients with breast cancer were initially assessed (Figure 1). We excluded 2735 patients who did not receive NAC. After excluding patients who achieved pCR, there were 183 patients left with a residual tumor. Patients who met the following criteria were further excluded.

(i)MRI obtained from outside the hospital or no assessable MRI;(ii)Discontinuation of therapy;(iii)Radiologic complete response;(iv)Stable disease or disease progression on MRI.

The following clinical data were collected: age at diagnosis, tumor size radiologically measured before NAC, radiologically measured extent of non-mass enhancement (NME) before NAC, lymph node metastasis, histological grade (HG), status of estrogen receptor (ER), progesterone receptor (PR), and HER2, surgery type (total mastectomy or breast conserving surgery), treatment modalities before and after surgery (chemotherapy regimen, radiotherapy, and endocrine therapy), peripheral blood markers [absolute neutrophil count (ANC), absolute lymphocyte count (ALC), neutrophil to lymphocyte ratio (NLR)] before NAC, and recurrence. Tumor HG was determined by the modified Scarff–Bloom–Richardson grading system [9].

All slides of resected specimens after NAC were reviewed by pathologist (YJC) and RCB class was assigned [6].

### 2.2. Immunohistochemistry and Molecular Subtyping

As in our previously published study [10], 3-µm-thick tissue sections were cut from formalin-fixed paraffin-embedded tissue blocks. After deparaffinization and rehydration with graded xylene and alcohol solutions, immunohistochemistry (IHC) was performed using a Ventana Discovery XT Automated Slide Stainer (Ventana Medical System, Tucson, AZ, USA). Cell Conditioning 1 buffer (citrate buffer, pH 6.0; Ventana Medical System) was used for antigen retrieval. The appropriate positive and negative controls were included.

Examination of IHC staining was performed using light microscopy (BX53 upright microscope, Olympus, Tokyo, Japan). Nuclear staining values of 1% or higher were considered indicative of ER (clone 6F11; dilution 1:200; Leica Biosystems, Wetzlar, Germany), PR (clone 16; dilution 1:500; Leica Biosystems) positivity [11]. HER2 (clone 4B5; dilution 1:5; Basel, Switzerland) staining was interpreted based on the 2018 American Society of Clinical Oncology/College of American Pathologists guidelines [12]. Only samples with strong and circumferential membranous HER2 immunoreactivity (3+) were considered positive, while those with 0 and 1+ HER2 staining were considered negative. Cases with equivocal HER2 expression (2+) were further evaluated for HER2 gene amplification by silver in situ hybridization. Positive nuclear Ki-67 (clone MIB; dilution 1:1000; Abcam, Cambridge, UK) staining was assessed based on the percentage of positive tumor cells, defined as the Ki-67 labeling index.

The specimens were categorized as follows:(i)Luminal/HER2-negative (ER- and/or PR-positive and HER2-negative);(ii)HER2-positive (HER2-positive regardless of ER and PR status);(iii)TNBC (ER-, PR-, and HER2-negative).

### 2.3. MRI Protocol and Image Analysis

To evaluate the tumor response to neoadjuvant chemotherapy, pre-neoadjuvant and post-neoadjuvant/preoperative breast MRIs were performed. We performed MRI examinations using a 3.0-T system (Achieva; Philips Medical Systems, Best, The Netherlands; General Electric Medical Systems, Milwaukee, WI, USA) with a dedicated four-channel breast coil. All images were obtained with bilateral axial views. The routine protocol comprised turbo spin-echo T1-weighted (repetition time/echo time, 505/10 msec; matrix, 564 × 338; field of view, 20–34 cm; and slice thickness, 3 mm) and T2-weighted, fat suppressed, spin echo sequences (repetition time/echo time, 5506/70 ms; matrix, 564 × 261; field of view, 20–34 cm; slice thickness, and 3 mm). Dynamic contrast-enhanced MR examination was performed with one pre-contrast and five post-contrast series using a T1-weighted gradient echo sequence (repetition time/echo time, 5/2.5 ms; matrix, 340 × 274; flip angle, 12°; field of view, 34 cm; and slice thickness, 2 mm), followed by image subtraction. Dynamic contrast-enhanced MRIs were obtained with injection of Gadobutrol (Gadovist, Bayer Healthcare, Berlin, Germany) with a dose of 0.1 mmol/kg, using an automated injector (Nemoto; Nemoto Kyorindo, Tokyo, Japan) at a rate of 2 mL/s and a 20 mL saline flush.

The images were assessed by a radiologist (NLE) with 6 years of experience in breast imaging. The size of tumor and NME was measured as the maximum diameter measured on contrast-enhanced T1-weighted images of pretreatment and post-treatment MRI. On post-treatment MRI, the RTP in radiologic partial response was divided as the following (Figure 2):(i)Concentric (concentric shrinkage of initial tumor bed, or decrease to small foci);(ii)Scattered (diffusely decrease or decrease in intensity alone with maintenance of initial tumor bed).

Further, change of tumor mass size and extent of NME before and after NAC were assessed. The radiologist was blinded to the clinicopathologic results.

### 2.4. Statistical Analysis

Recurrence free survival (RFS) was defined as the period from primary curative surgery to the date of any recurrence (loco-regional or distant metastasis), death due to any cause, or the last follow-up. Distant metastasis-free survival (DMFS) was defined as the time between primary curative surgery and the diagnosis of breast cancer-derived distant metastasis, death due to any cause, or end of follow-up.

To compare the characteristics between the two groups, the Student’s *t*-test was used for continuous variables, and the chi-square or Fisher’s exact test was used for categorical variables. Univariate analysis of RFS and DMFS was performed using the Cox proportional hazard model. Survival curves were obtained by the Kaplan–Meier method, and two-group comparisons were performed by a log-rank test. Statistical analyses were performed using SPSS version 24 (IBM Corp., Armonk, NY, USA). The threshold for statistical significance was set at *p* < 0.05, with a 95% confidence interval (CI) not including 1.

## 3. Results

### 3.1. Baseline Characteristics According to the Degradation Pattern

A total of 91 patients were finally enrolled for analysis. The median age was 47 years (range, 26–75 years). Concentric and scattered patterns were observed in 50 (54.9%) and 41 (45.1%) patients, respectively. Clinical characteristics were compared based on the RTP (Table 1). A scattered pattern was associated with a larger extent of pre-treatment NME and higher rate of total mastectomy. There were no statistical differences with regard to the baseline characteristics, including age, ER, PR, HER2, HG, tumor size, clinical lymph node status, type of axillary surgery, RCB class, ANC, ALC, NLR, and the treatment modalities of breast cancer.

### 3.2. Prognostic Significance of RTP

At a median follow-up time of 37 months (range, 6–116 months), 11 patients had developed recurrence. Among them, 10 had distant metastases. There were only two mortality events. RTP was not significantly associated with RFS [Figure 3a; hazard ratio (HR), 0.8113, 95% CI 0.259–2.877, *p* = 0.8113, log rank test], or DMFS (Figure 3b; HR 0.5761, 95% CI 0.197–2.471, *p* = 0.5761, log rank test). RTP was not a significant factor for the prediction of RFS or DMFS (Table 2).

## 4. Discussion

This study investigated the clinical characteristics and survival outcomes according to different RTP after NAC in patients with residual breast cancer. To the best of our knowledge, this is the first study focused on the patients with radiologic partial response who did not achieve pCR. Between the two different RTP groups (concentric and scattered), there was no difference in the baseline characteristics, except for the extent of pre-treatment NME. More importantly, survival outcomes between different RTPs were not significantly different (RFS, *p* = 0.8113; DMFS, *p* = 0.5761, log rank test).

Currently, NAC is widely applied for patients with locally advanced breast cancer. Even in early breast cancer, particularly HER2 and TNBC types, pCR serves as a prognostic indicator, and NAC enables the de-escalation of adjuvant treatment in those who achieve pCR. Regarding the NAC response, patients with a TNBC or HER2 tumor are more likely to have a significant reduction of the tumor and higher pCR rates compared to the patients with luminal breast cancer [13,14]. However, ironically, long-term survival is superior in luminal breast cancer, in contrast to the low treatment response rate. As breast cancer is a heterogeneous disease intratumorally and intertumorally [15], the response to NAC can also be varied.

A concentric pattern on the MRI has been considered as an effective predictor of pCR [16]. Additionally, different RTPs were also believed to reflect tumor biology because different molecular subtypes demonstrated different RTPs [16]. Pretreated TNBC often shows a well-defined mass and regressed in a concentrically shrinking pattern during NAC, whereas luminal breast cancer has no dominant radiologic morphology at the baseline or after NAC, and tends to maintain the initial extent of the tumor bed even when partial regression is obtained. This difference might be explained by the tumor heterogeneity, particularly in the luminal type of tumors, for which the NAC response rate is low [16].

Previous studies had focused on the achievement of pCR with regard to different RTPs on the MRI. Goorts et al. observed the RTP of 80 breast tumors halfway through NAC and reported that the shrinkage pattern during NAC could more accurately predict pCR [17]. Ballesio et al. investigated 51 breast tumors and found the association of concentric pattern-HER2 type-pCR and mixed pattern-luminal type- residual disease [18]. Interestingly, Fukada et al. analyzed the prognosis of low-grade luminal breast cancer after NAC based on the RTP [5]. An excellent DFS and OS were observed in tumors with a concentric pattern in both the developmental and validation sets [5]. The main difference between the previous studies and our study might be the patient cohort for analysis—we excluded pCR cases at first, as well as the radiologically stable disease, progressive disease, and cases with a complete response; cases with radiologic partial response alone were selected. In our study, we expected that different RTPs might show different clinicopathologic characteristics, however, no differences were found in the tumor characteristics or clinical outcomes.

Our study had a few limitations. First, it was a retrospective study with a small sample size, a short follow-up time, and the enrolled patients received uncontrolled treatments. There was a higher proportion of the TNBC subtype in our study than that observed in the real world, which could have led to selection bias. Second, our study had limited information about proliferative marker indices such as the Ki-67 labeling index and tumor infiltrating lymphocytes (TILs) in pre-treated biopsy tissue [19,20]. Both TILs and the Ki-67 labeling index are significant predictive markers for the NAC response. However, as our hospital is a tertiary referral institute, the slides were returned to the referring hospitals after review. Thus, a thorough pathologic review of the pre-treated biopsy slides was difficult. Furthermore, for older data dated before 2016, the pathologic evaluation of treated specimens was also limited—there was no information on proliferative markers, TILs, or the residual cancer burden index. Despite these limitations, our study revealed that there was no difference in the baseline characteristics and survival outcomes based on RTP in patients with radiologic partial response, excluding pCR.

## 5. Conclusions

In conclusion, radiologically different RTP has no prognostic effect in breast cancer patients receiving NAC. Further prospective studies are needed to validate the survival outcome according to the RTP in breast cancer patients with residual disease after NAC.

## Figures and Tables

**Figure 1 diagnostics-12-02294-f001:**
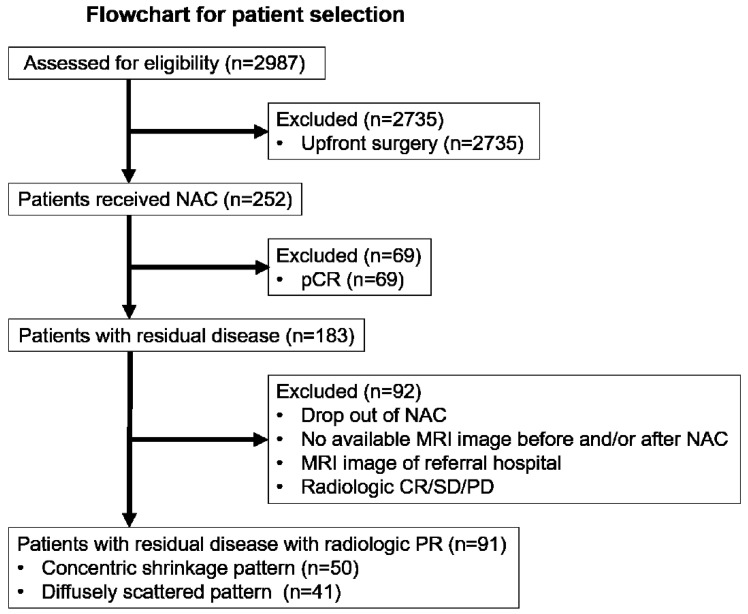
Flowchart for patient selection. NAC, neoadjuvant chemotherapy; PR, partial response; CR, complete response; SD, stable disease; PD, progressive disease.

**Figure 2 diagnostics-12-02294-f002:**
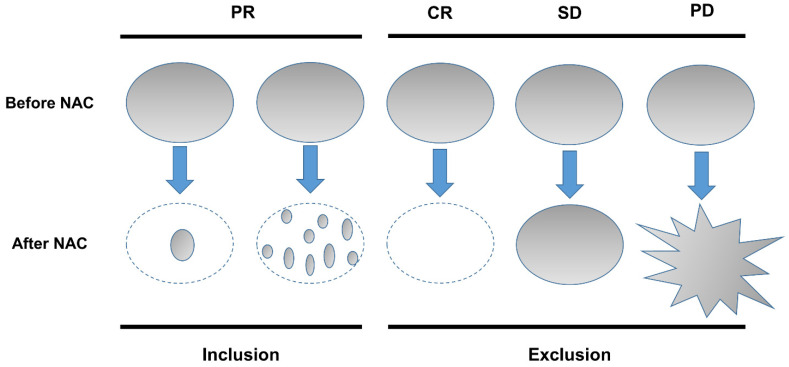
Magnetic resonance imaging (MRI)-based response patterns of breast tumors on breast MRI before and after neoadjuvant chemotherapy. PR, partial response; CR, complete response; SD, stable disease; PD, progressive disease; NAC, neoadjuvant chemotherapy.

**Figure 3 diagnostics-12-02294-f003:**
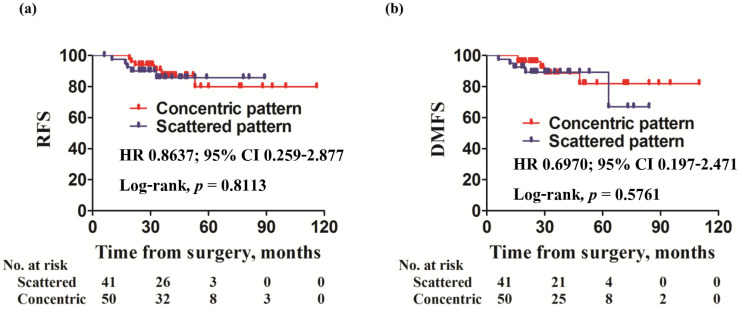
Kaplan–Meier survival curves of recurrence free survival (RFS) and distant metastasis-free survival (DMFS) according to tumor degradation pattern. No significant difference was found in RFS or DMFS based on the residual tumor pattern ((**a**), RFS, HR 0.8637, 95% CI 0.259–2.877, *p* = 0.8113; (**b**), DMFS, HR 0.6970, 95% CI 0.197–2.471; *p =* 0.5761, log-rank test, respectively). RFS, recurrence-free survival; DMFS, distant metastasis-free survival.

**Table 1 diagnostics-12-02294-t001:** Clinical characteristics in relation to tumor degradation pattern.

	Concentric, n = 50 (%)	Scattered, n = 41 (%)	*p* Value
Age (year, mean ± SD)	46.54 ± 9.89	49.17 ± 11.66	0.247
ER ^a^			0.466
Positive	24 (48)	24 (58.5)	
Negative	22 (44)	16 (39.0)	
Missing	4 (8)	1 (2.4)	
PR ^a^			0.817
Positive	23 (46)	19 (46.3)	
Negative	23 (46)	21 (51.2)	
Missing	4 (8)	1 (2.4)	
HER2 ^a^			0.260
Positive	8 (16)	11 (26.8)	
Negative	38 (76)	29 (70.7)	
Missing	4 (8)	1 (2.4)	
HG ^a^			0.295
I, II	13 (26)	12 (29.3)	
III	8 (16)	3 (7.3)	
Missing	29 (58)	26 (63.4)	
Subtype ^a^			0.486
Luminal/HER2(−)	23 (46)	19 (46.3)	
HER2 (+)	8 (16)	11 (26.8)	
TNBC	15 (30)	10 (24.4)	
Missing	4 (8)	1 (2.4)	
Tumor size before neoadjuvant treatment ^a^ (cm, mean ± SD)	3.59 ± 1.62	4.05 ± 2.96	0.374
NME size before neoadjuvant treatment ^a^ (cm, mean ± SD) ^a^	0.57 ± 1.87, n = 47	2.48 ± 3.43	0.002
Clinical lymph node metastasis			0.374
Negative	4 (8)	1 (2.4)	
Positive	46 (92)	40 (97.6)	
Breast surgery			<0.001
Breast conserving surgery	35 (70)	9 (22.0)	
Total mastectomy	15 (30)	32 (78.0)	
Axillary surgery			0.252
SLNB	15 (30)	8 (19.5)	
ALND	35 (70)	33 (80.5)	
ANC (cells/mm^3^, mean ± SD)	4170 ± 1610	4030 ± 1770	0.702
RCB class			0.687
I	4 (4.4)	2 (2.2)	
II/III	46 (50.5)	39 (42.9)	
ALC (cells/mm^3^, mean ± SD)	1810 ± 520	1830 ± 770	0.902
NLR (mean ± SD)	2.64 ± 1.77	2.51 ± 1.94	0.751
Anthracycline regimen			0.469
Included	42 (84)	32 (78.0)	
Not included	8 (16)	9 (22.0)	
Taxane regimen			0.801
Included	47 (94)	38 (92.7)	
Not included	3 (6)	3 (7.3)	
Radiotherapy			0.587
Done	49 (98)	39 (95.1)	
Not done/Unknown	1 (2)	2 (4.9)	
Endocrine therapy			0.533
Done	26 (52)	24 (58.5)	
Not done/Unknown	24 (48)	17 (41.5)	

SD, standard deviation; ER, estrogen receptor; PR, progesterone receptor; HER-2, human epidermal growth factor receptor-2; HG, histological grade; TNBC, triple negative breast cancer; NME, non-mass enhancement; SLNB, sentinel lymph node biopsy; ALND, axillary lymph node dissection; ANC, absolute neutrophil count; ALC, absolute lymphocyte count; NLR, neutrophil to lymphocyte ratio. ^a^ Percentages calculated without missing values.

**Table 2 diagnostics-12-02294-t002:** Univariate analysis for recurrence-free survival (RFS) and distant metastasis-free survival (DMFS).

	RFS		DMFS	
	HR (95% CIs)	*p* Value	HR (95% CIs)	*p* Value
Age	0.956 (0.900–1.016)	0.147	0.965 (0.908–1.026)	0.257
ER		0.120		0.219
Negative	1		1	
Positive	0.376 (0.110–1.290)		0.452 (0.127–1.603)	
PR		0.346		0.397
Negative	1		1	
Positive	0.478 (0.103–2.218)		0.511 (0.108–2.416)	
HER2		0.228		0.206
Negative	1		1	
Positive	0.279 (0.035–2.227)		0.258 (0.032–2.104)	
Subtype		0.256		0.811
Luminal/HER2(−)	1		1	
HER2 (+)	0.366 (0.042–3.187)		0.308 (0.035–2.703)	
TNBC	1.996 (0.568–7.012)		1.714 (0.447–6.573)	
HG		0.396		0.321
I, II	1		1	
III	1.306 (0.705–2.418)		1.086 (0.551–2.142)	
Tumor size before neoadjuvant treatment	1.181 (0.961–1.451)	0.114	1.177 (0.958–1.447)	0.121
NME size before neoadjuvant treatment	0.534 (0.195–1.464)	0.222	0.532 (0.191–1.486)	0.229
Clinical lymph node metastasis		0.790		0.732
Negative	1		1	
Positive	0.756 (0.096–5.948)		0.697 (0.088–5.529)	
Breast surgery		0.121		0.161
Breast conserving surgery	1		1	
Total mastectomy	0.350 (0.093–1.320)		0.380 (0.098–1.472)	
Axillary surgery		0.606		0.325
SLNB	1		1	
ALND	1.497 (0.323–6.943)		2.827 (0.357–22.401)	
ANC	1.123 (0.841–1.500)	0.430	1.124 (0.821–1.539)	0.467
ALC	0.360 (0.104–1.248)	0.107	0.273 (0.073–1.023)	0.054
NLR	1.154 (0.949–1.403)	0.152	1.178 (0.964–1.441)	0.110
Anthracycline regimen		0.458		0.303
Not included	1		1	
Included	0.604 (0.159–2.290)		0.481 (0.119–1.936)	
Taxane regimen		0.789		0.874
Not included	1		1	
Included	0.755 (0.097–5.900)		0.845 (0.106–6.755)	
Radiotherapy		0.266		0.349
Not done	1		1	
Done	0.309 (0.039–2.446)		0.370 (0.046–2.970)	
Endocrine therapy		0.088		0.136
Not done	1		1	
Done	0.313 (0.082–1.187)		0.356 (0.092–1.384)	
Degradation pattern		0.812		0.578
Concentric	1		1	
Scattered	1.156 (0.351–3.804)		1.423 (0.410–4.945)	

HR, hazard ratio; 95% CIs, 95% confidence intervals; ER, estrogen receptor; PR, progesterone receptor; HER-2, human epidermal growth factor receptor-2; HG, histological grade; TNBC, triple negative breast cancer; NME, non-mass enhancement; SLNB, sentinel lymph node biopsy; ALND, axillary lymph node dissection; ANC, absolute neutrophil count; ALC, absolute lymphocyte count; NLR, neutrophil to lymphocyte ratio.

## Data Availability

Availability of data and materials. All data generated or analyzed during this study are included in this research article and its additional files.

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
