# Peer review of "Residual Tumor Patterns of Breast Cancer on MRI after Neo-Adjuvant Chemotherapy: Impact on Clinicopathologic Factors and Prognosis"

_diagnostics, 2022, doi:10.3390/diagnostics12102294_

Round 1
Reviewer 1 Report
I congratulate the authors for the work that appears well designed and well written. it is appropriate to indicate, as required by the Helsinky Declaration, that the study, albeit retrospective, has the approval of the local ethics committee (indicating the protocol/date).
it would be advisable to specify in the methods with which timing (pre-post NAC) the MRIs were performed.
line 219 should be integrated with the concept that more and more NAC is used for operable HER2 / TNBC diseases as a function of recent data on deescalation / escalation depending on the pCR or non-pCR.
line 226: delete the comma
Author Response
Reviewer 1
I congratulate the authors for the work that appears well designed and well written. it is appropriate to indicate, as required by the Helsinky Declaration, that the study, albeit retrospective, has the approval of the local ethics committee (indicating the protocol/date).
[Response]
Thank you for your valuable comment. As you pointed out, we added the Declaration of Helsinki to the ‘Institutional Review Board statement’, and also added the date of IRB approval as below (lines 269-274)
Institutional Review Board Statement: The study protocol was approved by the institutional re-view board (IRB) of the Gangnam Severance Hospital (local IRB No.3-2021-0073). The need for informed consent was waived under the approval of the IRB due to the retrospective study design.
à Institutional Review Board Statement: All procedures performed in studies involving human participants were in accordance with the ethical standards of the institutional and/or national research committee and with the 1964 Declaration of Helsinki and its later amendments or comparable ethical standards. The study protocol was approved by the institutional review board (IRB) of the Gangnam Severance Hospital (local IRB No.3-2021-0073/2021-04-12). The need for in-formed consent was waived under the approval of the IRB due to the retrospective study design.
It would be advisable to specify in the methods with which timing (pre-post NAC) the MRIs were performed.
[Response]
Thank you for your valuable comment. As you pointed out, we added these sentences to Methods. (lines 113-114)
2.3. MRI protocol and image analysis
“To evaluate the tumor response to neoadjuvant chemotherapy, pre-neoadjuvant and post-neoadjuvant/preoperative breast MRIs were performed.”
line 219 should be integrated with the concept that more and more NAC is used for operable HER2 / TNBC diseases as a function of recent data on deescalation / escalation depending on the pCR or non-pCR.
[Response]
Thank you for your comment. As you mentioned, NAC in early breast cancer, especially HER2 and TNBC tumors, is considered more effective treatment nowadays. In this setting pCR could be the indicator of patient prognosis and support the de-escalation of adjuvant CTx. Even patient could not achieve pCR with NAC, subsequent appropriate adjuvant CTx is recommended. We further integrated this concept in the manuscript. (lines 209-211)
line 226: delete the comma
[Response]
Thank you for your comment. As you pointed out, we removed duplicate comma. (line 224)
We thank the reviewer for your careful reading of our manuscript and your helpful comments and suggestions, which were valuable and helpful for revising and improving our manuscript. We now hope that our revisions meet your requirements for publication.
Sincerely,
Chang Ik, Yoon.

Reviewer 2 Report
This study aims at comparing the clinical outcome of breast cancer patients with different residual tumor patterns on breast MRI after neoadjuvant chemotherapy.
The study is interesting, since neoadjuvant chemotherapy is increasingly used in clinical practice, and knowing the prognostic implications of response patterns might be important in patient management. The follow-up time is short in this study.
I would like to suggest some corrections and ask for some clarification.
Abstract: The abstract is quite clear, I suggest adding the time of follow-up considered in the study.
Introduction:
- Line 53-54 check for text clarity (eg “in effectively”)
Materials and methods:
- The patient’s flow diagram lacks a legend.
- Lines 82 to 89 cannot be regarded as a legend for figure 1:
o lines 82-84 do not correlate with the image and seem to refer to a comparison image of the tumor MRI-pattens, which is not present.
o lines 82-84 describe in poor English the patient flow diagram and are therefore redundant, can be eliminated
- Regarding the patient flow-chart I suggest some modifications:
o 1st box “assessed for eligibility criteria” à should be made explicit if they are all patients treated for breast cancer between 2009 e 2017
o 6th box from the top: check for English, in particular, “MRI image with external hospital”
o 7th box from the top “patients with non-pathologic complete response” appears superfluous, I suggest unifying all the exclusion criteria in one box, merging the 6th (“excluded for the following reasons …”) and the 8th box (“excluded patients with radiologic complete response …”).
- “As previously described”, line 91 à I don’t see where in the text
- Line 94 “respectively” appears superfluous
- Figure 2: is missing the acronym legend.
Results:
- Line 157-159 are from the template, and must be deleted.
- Table 1: check for % in ER-negative concentric pattern.
I suggest boldening significant p's and graphically dividing the table into sections (e.g., with thicker lines) to make the table clearer.
- Line 182 -183 appear more adequate for the materials and methods section.
- Table 2: I suggest boldening significant p's and graphically dividing the table into sections (e.g., with thicker lines) to make the table clearer.
I suggest the term breast surgery instead of operation.
- Figure 3: the images are blurry, please replace them with images with a better-quality definition
Discussion:
- Verbal concordances need to be reviewed
- The discussion needs to be better organized. I suggest ordering it by addressing the two most important findings: the non-difference in characteristics between tumors with the concentric and the scattered response, and the non-difference in survival outcomes. For this reason, I find redundant repeating several times in the text of the concept present in lines 207 and 208, it is necessary to compact all references to this result into one section of the text.
- Line 219-222 are more appropriate for the introduction, not in the discussions.
- Line 231: change term, “we revealed” does not fit in the text.
- In lines 229-231 it is stated that "So far, exploration of whether residual tumor pattern predicts recurrence after NAC and curative resection has been limited to patients who did not achieve pCR." which contradicts "To the best of our knowledge, this is the first study focused on patients with partial radiological response who did not achieve pCR." lines 203-205.
- Lines 233-244: better explore the characteristics of the cohorts of the cited studies and the follow-up time, which for example in the study by Fukada et al. is 67.9 months
- By removing some unnecessary and repetitive sections, the discussions need to be implemented with more references to the literature.
Conclusions:
- The conclusions are too brief and seem overly peremptory. Although this study is an interesting retrospective study, however, it has a limited cohort of patients, a short follow-up time, a retrospective design, and some results that deviate from those in the literature. It is necessary to at least add to the conclusions that studies with larger sample sizes and prospective studies are needed.
Author Response
Reviewer 2
This study aims at comparing the clinical outcome of breast cancer patients with different residual tumor patterns on breast MRI after neoadjuvant chemotherapy.
The study is interesting, since neoadjuvant chemotherapy is increasingly used in clinical practice, and knowing the prognostic implications of response patterns might be important in patient management. The follow-up time is short in this study.
I would like to suggest some corrections and ask for some clarification.
Abstract: The abstract is quite clear, I suggest adding the time of follow-up considered in the study.
[Response]
Thank you for your comment. As you pointed out, we added the median follow up period (37 months) in the result section of abstract. (line 27)
Introduction:
- Line 53-54 check for text clarity (eg “in effectively”)
[Response]
Thank you for your comment. We revised the sentence as below for clarity. (lines 53-54)
RCB classes are now established prognostic indicator that effectively stratify the patients after long-term follow up
Materials and methods:
- The patient’s flow diagram lacks a legend.
- Lines 82 to 89 cannot be regarded as a legend for figure 1:
o lines 82-84 do not correlate with the image and seem to refer to a comparison image of the tumor MRI-pattens, which is not present.
o lines 82-84 describe in poor English the patient flow diagram and are therefore redundant, can be eliminated
- Regarding the patient flow-chart I suggest some modifications:
o 1st box “assessed for eligibility criteria” à should be made explicit if they are all patients treated for breast cancer between 2009 e 2017
o 6th box from the top: check for English, in particular, “MRI image with external hospital”
o 7th box from the top “patients with non-pathologic complete response” appears superfluous, I suggest unifying all the exclusion criteria in one box, merging the 6th (“excluded for the following reasons …”) and the 8th box (“excluded patients with radiologic complete response …”).
[Response]
We revised the figure 1 more concisely and rewrite the legend (lines 83-85). Previously existed lines 82-84 were deleted.
- “As previously described”, line 91 à I don’t see where in the text
[Response] The sentence was revised as below (line 88)
As previously described [10], 3-µm-thick tissue sections were cut from formalin-fixed paraffin-embedded tissue blocks.
àAs in our previously published study [10], 3-µm-thick tissue sections were cut from formalin-fixed paraffin-embedded tissue blocks.
- Line 94 “respectively” appears superfluous
[Response] We deleted this word. (line 90)
- Figure 2: is missing the acronym legend.
[Response] We added acronym in legend of Figure 2. (lines 140-142)
Results:
- Line 157-159 are from the template, and must be deleted.
- Table 1: check for % in ER-negative concentric pattern.
[Response] Thank you for your careful comment. We deleted the template, and corrected typo in table 1 (48à 44).
I suggest boldening significant p's and graphically dividing the table into sections (e.g., with thicker lines) to make the table clearer.
[Response]
Thank you for your comment. As pointed out, the p value in the tables was changed to bold, and the variables in the tables were graphically divided. (Table 1 and 2)
- Line 182 -183 appear more adequate for the materials and methods section.
- Table 2: I suggest boldening significant p's and graphically dividing the table into sections (e.g., with thicker lines) to make the table clearer.
[Response]
Thank you for your comment.
We moved the lines 182-183 to the materials and methods section (lines 151-152). Same as above mentioned, the p value in the tables was changed to bold, and the variables in the tables were graphically divided. (Table 1 and 2)
I suggest the term breast surgery instead of operation.
[Response]
Thank you for your comment. As pointed out, we changed term from breast operation to breast surgery.
- Figure 3: the images are blurry, please replace them with images with a better-quality definition
[Response]
Thank you for your comment. As pointed out, we changed a better-quality figure.
Discussion:
- Verbal concordances need to be reviewed
- The discussion needs to be better organized. I suggest ordering it by addressing the two most important findings: the non-difference in characteristics between tumors with the concentric and the scattered response, and the non-difference in survival outcomes. For this reason, I find redundant repeating several times in the text of the concept present in lines 207 and 208, it is necessary to compact all references to this result into one section of the text.
[Response]
Thank you for your comment. We totally agreed to your comment. We revised the discussion section for better organization.
- Line 219-222 are more appropriate for the introduction, not in the discussions.
[Response]
Thank you for your comment. As same content with same reference is present in the introduction, we deleted the sentence.
- Line 231: change term, “we revealed” does not fit in the text.
[Response]
Thank you for your comment. During revising the discussion section, that sentence was deleted to reduce redundancy.
- In lines 229-231 it is stated that "So far, exploration of whether residual tumor pattern predicts recurrence after NAC and curative resection has been limited to patients who did not achieve pCR." which contradicts "To the best of our knowledge, this is the first study focused on patients with partial radiological response who did not achieve pCR." lines 203-205.
[Response]
The sentence “So far, exploration of whether residual tumor pattern predicts recurrence after NAC and curative resection has been limited to patients who did not achieve pCR.” seems to be wrongly inserted. We deleted the sentence. Thank you for your careful comment.
- Lines 233-244: better explore the characteristics of the cohorts of the cited studies and the follow-up time, which for example in the study by Fukada et al. is 67.9 months
[Response]
Thank you for your comment. We further described the characteristics of the cohort of cited studies.
- Goorts et al: lines 226-227
- Ballesio et al: lines 228-229
- Fukada et al: line 230-232
- By removing some unnecessary and repetitive sections, the discussions need to be implemented with more references to the literature.
[Response]
Thank you for your comment. We further revised the discussion section.
Conclusions:
- The conclusions are too brief and seem overly peremptory. Although this study is an interesting retrospective study, however, it has a limited cohort of patients, a short follow-up time, a retrospective design, and some results that deviate from those in the literature. It is necessary to at least add to the conclusions that studies with larger sample sizes and prospective studies are needed.
[Response]
Thank you for your comment. We further mentioned the limitations and revised the conclusion more carefully.
We thank the reviewer for your careful reading of our manuscript and your helpful comments and suggestions, which were valuable and helpful for revising and improving our manuscript. We now hope that our revisions meet your requirements for publication.
Sincerely,
Chang Ik, Yoon.
Round 2
Reviewer 2 Report
I thank the authors for the thorough corrections. The article can be accepted in present form.
Sincerely